# Association between prediagnosis depression and mortality among postmenopausal women with colorectal cancer

**Xiaoyun Liang** [1]*, **Michael Hendryx**[2], **Lihong Qi**[3], **Dorothy Lane**[4], **Juhua Luo**[5]

**1** School of Social Development and Public Policy, Beijing Normal University, Beijing, China, **2** Department of Environmental and Occupational Health, School of Public Health, Indiana University Bloomington, Bloomington, Indiana, United States of America, **3** Department of Public Health Sciences, University of California Davis, Davis, CA, United States of America, **4** Department of Family, Population and Preventive Medicine, Stony Brook University, School of Medicine, Stony Brook, New York, United States of America, **5** Dept. of Epidemiology and Biostatistics, School of Public Health, Indiana University Bloomington, Bloomington, Indiana, United States of America

* liangxiaoyun@bnu.edu.cn

## Abstract

### Background

There are no epidemiologic data on the relation of depression before colorectal cancer diagnosis to colorectal cancer mortality among women with colorectal cancer, especially those who are postmenopausal. Our aim was to fill this research gap.

### Methods

We analyzed data from a large prospective cohort in the US, the Women's Health Initiative (WHI). The study included 2,396 women with incident colorectal cancer, assessed for depressive symptoms and antidepressant use before cancer diagnosis at baseline (screening visit in the WHI study) during 1993–1998. Participants were followed up from cancer diagnosis till 2018. We used Cox proportional hazards regression to estimate adjusted hazard ratios (HRs) between depression (depressive symptoms or antidepressant use) at baseline, and all-cause mortality and colorectal cancer-specific mortality.

### Results

Among women with colorectal cancer, there was no association between baseline depression and all-cause mortality or colorectal cancer-specific mortality after adjusting for age or multiple covariates.

### Conclusion

Among women with colorectal cancer, there was no statistically significant association between depression before colorectal cancer diagnosis and all-cause mortality or colorectal cancer-specific mortality. Further studies are warranted to assess depressive symptoms and antidepressant use, measured at multiple points from baseline to diagnosis, and their interactions with specific types of colorectal cancer treatment on the risk of death from colorectal cancer.

**Data Availability Statement:** Data are available from the WHI upon reasonable request (www.whi.org), and please contact helpdesk@whi.org.

**Funding:** The WHI program is funded by the National Heart, Lung, and Blood Institute, National Institutes of Health, US Department of Health and Human Services, through contracts HHSN268201100046C, HHSN268201100001C, HHSN268201100002C, HHSN268201100003C, HHSN268201100004C, and HHSN271201100004C. Research reported in this publication was supported by the National Cancer Institute of the National Institutes of Health under Award Number R15CA179463 (Juhua Luo). The content is solely the responsibility of the authors and does not necessarily represent the official views of the National Institutes of Health.

**Competing interests:** The authors have declared that no competing interests exist.

**Abbreviations:** AMM, Antidepressant Medications Management; BMI, body mass index; CES-D, Center for Epidemiologic Studies Depression; CI, confidence interval; CRC, colorectal cancer; CT, Clinical Trial; DIS, Diagnostic Interview Schedule; MDDB, Master Drug Database; NDC, National Drug Code; NSAIDs, nonsteroidal antiinflammatory drugs; OS, Observational Study; HRs, Hazard ratios; RR, relative risk; SD, standard deviation; IARC, International Agency for Research on Cancer; SSRIs, selective serotonin reuptake inhibitors; WHI, Women's Health Initiative.

# Introduction

According to International Agency for Research on Cancer (IARC) estimates, in 2018, colorectal cancer accounted for 9.5% of all incident cancers among women in the world, making it the second most common cancer in women [1]. In 2020 in the United States, it is estimated that 69,650 women will be diagnosed with colorectal cancer and 24,570 will die from it [2].

The effect of depression or antidepressant use on colorectal cancer risk has been of interest in some studies [3–7]. These studies showed that severe depressive symptoms were associated with an increased risk of colorectal cancer [3], and that high daily dose of selective serotonin reuptake inhibitors (SSRIs) was associated with a decreased risk of colorectal cancer [5]. Several studies have also evaluated the relationship between depression after colorectal cancer diagnosis and its prognosis [8, 9]. Richardson et al. measured depression among 47 rectal cancer patients after surgery, and found no association between depression and survival [8]. Schofield et al. followed up 429 subjects with metastatic colorectal cancer for an average of 31 months, and suggested that depression after cancer diagnosis was associated with poor survival in patients with advanced colorectal cancer [9]. Worse adherence to medical advice and self-care is a speculative mechanism by which depression might affect survival. A review summarized the relationship between depression before cancer diagnosis and cancer mortality [10], including breast, leukemia/lymphoma, lung and brain cancer, but not colorectal cancer, and found that depression diagnosis and severe depressive symptoms before cancer diagnosis predicted increased mortality. There are no epidemiologic data on the relation of depression before colorectal cancer diagnosis to colorectal cancer mortality among women with colorectal cancer.

Mechanisms between depression and cancer survival are not clear. It may be that depression affects the neuro-endocrine axis, neuro-immunological function, or other central nervous system processes [11]. Central nervous system dysregulation may place a person at higher risk for cancer incidence, progression and mortality [12]. A second possible mechanism is that people with depression might change behaviors (such as engaging in less physical activity and more smoking), be less likely to engage in social activity (resulting in less social support and social capital), or have less adherence to medical recommendations (such as preventive screening and medical treatment) [12, 13], which could lead to worse survival in cancer patients. A third possible mechanism is that cancer survivors have difficulties following chemotherapy and other forms of cancer treatment [12], which can lead to worse medical outcome. Therefore, depressive symptoms might influence the effect of conventional cancer medical treatment [12]. An animal experiment suggested that depression increased catecholamine levels and promoted liver metastasis of colon cancer [14].

The prevalence of depression among women is approximately twice that of men [15]. The Women's Health Initiative (WHI) is a large, ongoing prospective cohort study in the US with extensive data on exposures and potential confounders, and more than 20 years of follow-up. We utilized WHI data to assess the association between depression symptoms, antidepressant use and the risk of colorectal cancer mortality among women with colorectal cancer.

# Methods

## Participants

We used data from the WHI Observational Study (OS) and Clinical Trial (CT), a prospective study based on 40 clinical centers throughout the United States, in which 161,808 postmenopausal women aged 50 to 79 years old were enrolled [16]. Postmenopausal was defined as women aged 50–54 with no menstrual period for at least one year and women aged 55 or above with no menstrual period for at least half a year [17]. Women were included if they

intended to reside in the study area for 3 years or above, and excluded if they had a survival time of fewer than 3 years, or had poor adherence of follow-up such as having alcoholism or dementia, or participated in another clinical trial [16]. We included 2994 WHI OS and CT women newly diagnosed with invasive colorectal cancer in the analysis, but did not include 223 women with colorectal cancer determined solely by the cause of death without central adjudication because the date of cancer diagnosis was unknown.

Women were excluded if they had any prior diagnosis of cancer other than non-melanoma skin cancer before the date of the baseline questionnaire administration ($n$ = 341) ("Baseline" in the current study refers to the screening visit in the WHI study). Women with missing data on depressive symptoms at baseline ($n$ = 53), or cancer stage ($n$ = 1) were excluded. Women with missing covariate data, i.e. smoking, physical activity, body mass index, total energy intake, dietary fiber, percent calories from fat, or receipt of colonoscopy, were excluded ($n$ = 203). For those missing data on family history of colorectal cancer (n = 193), we created an indicator variable and included it in the multivariable model. A flowchart showing derivation of the included study population is presented as Fig 1.

The Women's Health Initiative was overseen by ethics committee at all 40 clinical centers (Albert Einstein College of Medicine, Baylor College of Medicine, Brigham and Women's Hospital, Harvard University, Brown University, Emory University, Fred Hutchinson Cancer Center, George Washington University Medical Center, Harbor-UCLA Research and Education Institute, Kaiser Permanente Center for Health Research [Portland, OR], Kaiser Permanente Division of Research [Oakland, CA], Medical College of Wisconsin, Howard University, Northwestern University, Rush-Presbyterian St. Luke's Medical Center, Stanford Prevention Research Center, State University at Stony Brook, Ohio State University, University of Arizona, University of Buffalo, University of California-Davis, University of California-Irvine,

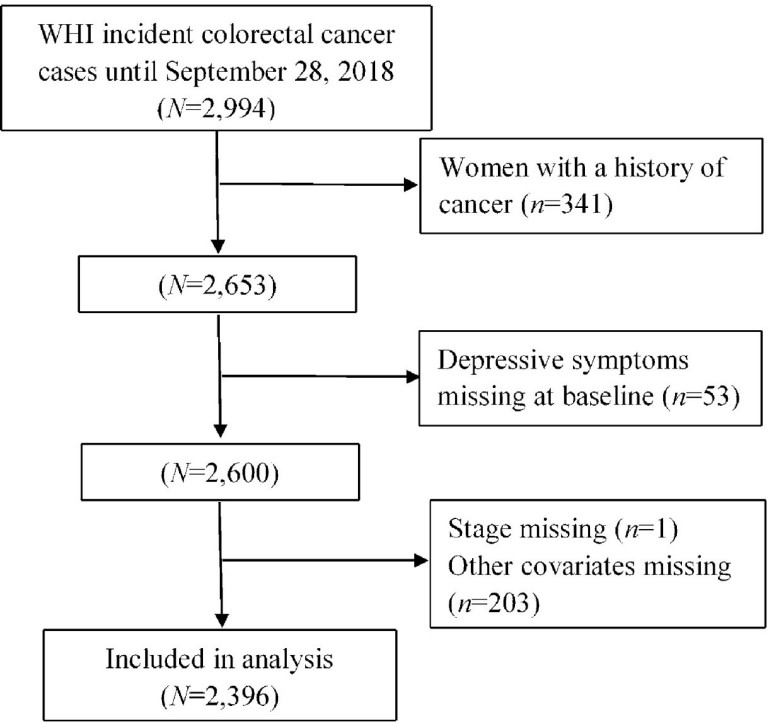

**Fig 1. Flow diagram of participants included in the analysis.**

University of California-Los Angeles, University of California-San Diego, University of Cincinnati, University of Florida, University of Hawaii, University of Iowa, University of Massachusetts, University of Medicine and Dentistry of New Jersey, University of Miami, University of Minnesota, University of Nevada, University of North Carolina-Chapel Hill, University of Pittsburgh, University of Tennessee, University of Texas, University of Wisconsin-Madison, Wake Forest University School of Medicine, Wayne State University School of Medicine), by the coordinating center (Fred Hutchinson Cancer Center), and an independent data and safety monitoring board for the clinical trials. Each institution obtained human subjects committee approval. Each participant provided written informed consent.

## Exposures

As described in detail previously [18], participants completed a questionnaire on depressive symptoms at baseline through in-person interview during 1993–1998. The depression scores were calculated from a short (8-item) form of the Center for Epidemiologic Studies Depression (CES-D) Scale. The range of depression score was 0–1. A higher score means a greater probability of depression. Depressive symptoms were classified as no/yes based on a previously established cutoff of 0.06 [19].

Women were asked to bring their original pill bottles for all medications to the baseline interview. Depression treatment was assessed from these medications using drug information from the Master Drug Database (MDDB; Medi-Span, Indianapolis, IN). We utilized the National Drug Code (NDC) for Antidepressant Medications Management (AMM) to classify whether the medication was an antidepressant or not [20]. In the analysis, we did not consider trazodone as an antidepressant because it was commonly used as a sedative or hypnotic [21]. Women were classified as antidepressant users and nonusers.

Consistent with our prior study in the WHI [18], a variable "depression" combined depressive symptoms and antidepressant use, which categorized women with either depressive symptoms or antidepressant use as "depressed" and those with neither variable as "non-depressed".

## Outcomes

The diagnosis of colorectal cancer was identified via self-reported questionnaires, verified by physicians with medical records and pathology reports, and locally adjudicated by a Clinical Center and then centrally by the WHI Clinical Coordinating Center [22]. The outcomes in this study were all-cause mortality and colorectal cancer-specific mortality. All causes of death were centrally adjudicated by physicians utilizing medical records, death certificates, and informant interviews [22]. Women who were alive were censored at the end of follow-up time based on the most recent WHI data update (September 28, 2018).

Survival time was calculated as the date from colorectal cancer diagnosis until death or the last follow-up, whichever came first. For analyses of all-cause mortality, women alive at the last follow-up were administratively censored. For colorectal cancer-specific mortality, women who were alive at the last follow-up or who died from non-cancer causes were administratively censored.

## Covariates

Baseline covariates in WHI study, including demographic and lifestyle data as well as data about family and medical history, were acquired by self-reported questionnaires during 1993–1998. Age at baseline was grouped as <60, 60-<70, or ≥70 years old (<60 years old as the reference). Race/ethnicity was classified as non-Hispanic white or other. Some literature showed that persons with different races had various level of mortality [23, 24], and our data supported

that race was associated with depression so we considered race as a potential confounder in the analysis. Body mass index (BMI) was calculated as weight divided by height squared (kg/m$^2$), and classified as <25, 25-<30, or ≥30 (<25 as the reference). For smoking status, women were categorized as never smokers, former smokers and current smokers (never smokers as the reference). Physical activity was measured by weekly expenditure of energy from recreational physical activity including walking and other mild, moderate and strenuous exercise before the survey. Food and nutrient intake in the last 3 months were examined by a food frequency questionnaire. Physical activity, total energy intake, dietary fiber, and percent calories from fat were continuous variables. Consistent with our prior study [18], a comorbidity index was generated by combining 10 co-occurring self-reported conditions at baseline (non-cancer chronic diseases such as glaucoma, asthma, cardiovascular disease and hypertension), and was categorized as 0 (no comorbidity besides colorectal cancer), 1 (one comorbidity), 2 (two comorbidies) and 3 (three or more comorbidies). Family history of colorectal cancer mortality was categorized as yes, no, or unknown (no as the reference). Receipt of colonoscopy was classified as ever or never and NSAIDs use was categorized as yes or no. History of postmenopausal hormone use (estrogen-alone or estrogen plus progestin) was categorized as ever or never. Colorectal cancer stage was characterized as in situ, localized, regional/distant stage, or unknown (localized stage as the reference). Treatment arm in each CT was categorized as not enrolled, intervention group, or control group (not enrolled as the reference).

## Statistical analysis

We compared demographic and lifestyle variables and tumor characteristics between women with and without depression at baseline. Mean ± standard deviation was utilized to describe continuous characteristics, while proportion was utilized for categorical variables. T-test was used to test group differences for continuous variables, and Chi-square test for categorical variables.

For all-cause mortality, stratified Cox proportional hazards regression was used to meet the proportional hazard assumption. Adjusted hazard ratios (HRs) and 95% confidence intervals (CIs) for all-cause mortality were estimated using multivariable Cox proportional hazards regression stratified by age at baseline (<60, 60-<70, ≥70 years old), BMI (<25, 25-<30, ≥30), NSAIDs use (yes/no) and stage (in situ, localized, regional/distant, unknown). Adjusted HRs and 95% CIs for colorectal cancer-specific mortality were estimated using multivariable Cox proportional hazards regression. In the Cox models, potential confounders included: race/ethnicity, smoking, physical activity, total energy intake, dietary fiber, percent calories from fat, comorbidity at baseline, family history of colorectal cancer mortality, receipt of colonoscopy, history of postmenopausal hormone use, and treatment arm in each CT. Additional covariates such as age atbaseline, BMI, NSAIDs use, and tumor stage were included in the Cox model for colorectal cancer-specific mortality.

Effect modification analyses explored whether associations differed by tumor stage (localized, regional/distant) at diagnosis.

Three sensitivity analyses were carried out. First, we added the duration from baseline to colorectal cancer diagnosis into analysis models. Second, we categorized depressive symptoms as no/yes based on a cutpoint of 0.009. Last, we included only incident colorectal cancer diagnosed after one-year follow up to account for depression related to as-yet undiagnosed colorectal cancer.

## Results

Of the total 2,396 women, the prevalence of depression based on depressive symptoms or antidepressant use at baseline was 14.9% (*n* = 358). This included 10.8% (*n* = 259) based on

depressive symptoms and 5.7% (*n* = 137) based on antidepressant use, respectively (Table 1). Women with depression at baseline, compared to those without depression, were more likely to be younger, black, overweight/obese, physically inactive, have higher energy intake and percent calories from fat, have more comorbidities, and higher proportions of NSAIDs use and postmenopausal hormone use (all *p's*<0.05). For receipt of colonoscopy, duration from baseline to colorectal cancer diagnosis and tumor characteristics, there were no statistical differences between women with and without baseline depression.

Among women with colorectal cancer, there was no association of baseline depression and all-cause mortality or colorectal cancer-specific mortality after adjusting for age or multiple covariates (HR: 1.05, 95% CI: 0.89–1.23; HR: 0.90, 95% CI: 0.71–1.14 for all-cause mortality or colorectal cancer-specific mortality, respectively) (Table 2).

Effect modification analysis by tumor characteristics showed that there was no association of baseline depression and all-cause mortality or colorectal cancer-specific mortality among women with colorectal cancer in localized stage, or regional or distant stage (Table 3).

Duration from baseline to colorectal cancer diagnosis was 8.8 years on average. The results of three sensitivity analyses were consistent with the main findings when the follow-up duration from baseline to colorectal cancer diagnosis was added into analysis models, when depressive symptoms was categorized as no/yes based on a cutpoint of 0.009 instead of 0.06, or when only incident colorectal cancer diagnosed after one-year follow up was included.

## Discussion

The results of this prospective observational study indicate no statistically significant association between depression before colorectal cancer diagnosis on all-cause mortality or colorectal cancer-specific mortality.

Pinquart and Duberstein summarized 37 prospective studies and showed that depression or higher levels of depressive symptoms before cancer diagnosis increased cancer mortality (RR: 1.14; 95% CI: 1.06–1.23) [10], but colorectal cancer mortality was not specified.

One explanation for our null finding is that there may truly be no association between depression before colorectal cancer diagnosis and colorectal cancer prognosis. However, misclassification of depression measurement may play a major role for the null finding. First, depression is based on self-report and not clinical diagnosis. However, among WHI participants, reliability and validity of the depression-screening method have been previously evaluated and shown to have acceptable sensitivity (74%) and specificity (87%) compared to clinician diagnosis of depression [25]. In addition, combining antidepressant use and depressive symptoms could likely improve the identification of depressed women. Second, depression was measured only at baseline. Failing to take any changes of depression over time into account may lead to a large degree of exposure misclassification. The bias may be more likely to be non-differential, which would make our results toward the null. However, the WHI includes a follow-up period over 20 years between the measurement of depression and mortality. Third, depression was measured at baseline rather than at cancer diagnosis. The remote measurement may dilute its role on cancer prognosis. Persons already depressed at baseline may not change after cancer diagnosis, but persons without depression at baseline who developed cancer and then became depressed would be misclassified as not depressed. In addition, women with depression were more likely to be younger at diagnosis. Younger cancer patients may have more aggressive treatment than older patients, and therefore may have better prognosis; even though age was adjusted in the regression model there may still be residual confounding effect by age. This better prognosis among younger women may counterbalance the adverse impact by being depressed. However, we carried out an analysis among women under

**Table 1. Characteristics of participants by depression at WHI baseline (*n* = 2,396).**

| Variable | Depression at WHI baseline | | |
|---|---|---|---|
| | Depression | No depression | P value |
| Total number of women | 358 | 2,038 | |
| **WHI baseline characteristics** | | | |
| Age at WHI baseline (*n*, %) | | | <0.001 |
| <60 | 107 (29.9) | 416 (20.4) | |
| 60-<70 | 155 (43.3) | 1,020 (50.1) | |
| ≥70 | 96 (26.8) | 602 (29.5) | |
| Race/ethnicity (*n*, %) | | | <0.01 |
| American Indian or Alaskan Native | 3 (0.8) | 4 (0.2) | |
| Asian or Pacific Islander | 5 (1.4) | 51 (2.5) | |
| Black or African-American | 46 (12.9) | 172 (8.4) | |
| Hispanic or Latino | 11 (3.1) | 42 (2.1) | |
| White, non-Hispanic-ethnicity | 284 (79.3) | 1,742 (85.5) | |
| Other | 9 (2.5) | 27 (1.3) | |
| Body mass index (*n*, %) | | | <0.001 |
| <25 | 83 (23.2) | 695 (34.1) | |
| 25-<29.9 | 127 (35.5) | 692 (34.0) | |
| ≥30 | 148 (41.3) | 651 (31.9) | |
| Smoking status (*n*, %) | | | >0.05 |
| Never smokers | 176 (49.2) | 1,009 (49.5) | |
| Former smokers | 148 (41.3) | 883 (43.3) | |
| Current smokers | 34 (9.5) | 146 (7.2) | |
| Physical activity (mean±SD, METs/wk) | 8.6±10.9 | 12.0±13.0 | <0.001 |
| Total energy intake (mean±SD, med serv/day) | 1757.1±769.4 | 1606.9±667.6 | <0.001 |
| Dietary fiber (mean±SD, med serv/day) | 16.2±7.2 | 15.7±7.0 | >0.05 |
| Percent calories from fat (mean±SD, med serv/day) | 35.2±7.9 | 32.9±8.2 | <0.001 |
| Comorbidity index (*n*, %) | | | <0.001 |
| 0 | 51 (14.3) | 439 (21.5) | |
| 1 | 81 (22.6) | 634 (31.1) | |
| 2 | 99 (27.7) | 508 (24.9) | |
| 3+ | 127 (35.5) | 457 (22.4) | |
| Family history of CRC (*n*, %)* | 67 (18.7) | 386 (18.9) | >0.05 |
| Receipt of colonoscopy (*n*, %) | 179 (50.0) | 950 (46.6) | >0.05 |
| NSAIDs use (*n*, %) | 81 (22.6) | 323 (15.9) | <0.01 |
| History of postmenopausal hormone use (*n*, %) | 203 (56.7) | 966 (47.4) | <0.01 |
| **Characteristics at CRC diagnosis** | | | |
| Age at CRC diagnosis (*n*, %) | | | <0.01 |
| <70 | 135 (37.7) | 627 (30.8) | |
| 70-<80 | 159 (44.4) | 915 (44.9) | |
| ≥80 | 64 (17.9) | 496 (24.3) | |
| Duration from WHI baseline to CRC diganosis (mean±SD, yrs) | 8.5±5.8 | 8.8±5.6 | >0.05 |
| Tumor stage (*n*, %) | | | >0.05 |
| In situ | 11 (3.1) | 63 (3.1) | |
| Localized | 156 (43.6) | 855 (42.0) | |
| Regional | 143 (39.9) | 815 (40.0) | |
| Distant | 39 (10.9) | 267 (13.1) | |
| Unknown | 9 (2.5) | 38 (1.9) | |

WHI: Women's Health Initiative; SD: standard deviation; CRC: colorectal cancer.

* Missing for 193 women.

**Table 2. Hazard ratios (HRs) and 95% confidence intervals (CIs) for all-cause mortality and colorectal cancer specific mortality in relation to depression at WHI baseline.**

| | *n* | Deaths | Age-adjusted | | Multivariable-adjusted* | |
|---|---|---|---|---|---|---|
| | | | *HR* | *95% CI* | *HR* | *95% CI* |
| *All-cause mortality* | | | | | | |
| Depression | | | | | | |
| No | 2,038 | 997 | 1 | | 1 | |
| Yes | 358 | 183 | 1.11 | 0.94–1.30 | 1.05 | 0.89–1.23 |
| Depressive symptoms | | | | | | |
| No | 2,137 | 1,047 | 1 | | 1 | |
| Yes | 259 | 133 | 1.10 | 0.92–1.32 | 1.04 | 0.86–1.25 |
| Antidepressant use | | | | | | |
| No | 2,259 | 1,113 | 1 | | 1 | |
| Yes | 137 | 67 | 1.09 | 0.85–1.39 | 1.04 | 0.80–1.35 |
| *Colorectal cancer mortality* | | | | | | |
| Depression | | | | | | |
| No | 2,038 | 518 | 1 | | 1 | |
| Yes | 358 | 81 | 0.92 | 0.73–1.17 | 0.90 | 0.71–1.14 |
| Depressive symptoms | | | | | | |
| No | 2,137 | 538 | 1 | | 1 | |
| Yes | 259 | 61 | 0.95 | 0.73–1.25 | 0.90 | 0.68–1.19 |
| Antidepressant use | | | | | | |
| No | 2,259 | 572 | 1 | | 1 | |
| Yes | 137 | 27 | 0.81 | 0.55–1.19 | 0.79 | 0.52–1.18 |

* Results were adjusted for age at baseline, race/ethnicity, body mass index (BMI), smoking, physical activity, total energy intake, dietary fiber, percent calories from fat, comorbidity at baseline, family history of colorectal cancer mortality, receipt of colonoscopy, NSAIDs use, history of postmenopausal hormone use, tumor stage, and treatment arm in each CT.

60 at baseline, and similar results were found as that in the main analysis. Lastly, the association between depression and mortality is influenced by the disease condition itself, including its severity, chronicity, and symptom type [26]. In the WHI, severely depressed women were not excluded, but they possibly were less likely to participate in the study. Therefore, depression scores might be present in a restricted range, which could result in limited generalizability of findings.

Study strengths include a large sample with a long prospective follow-up period and adjudicated cancer diagnosis and cause of death data. Besides limitations identified above about depression, such as self-reported depression and only one measurement on depression at baseline, an additional limitation is that there is no treatment information for colorectal cancer in the WHI data.

## Conclusions

Among women with colorectal cancer, there is no statistically significant association between depression before colorectal cancer diagnosis and all-cause mortality or colorectal cancer-specific mortality. Next research steps could include examining the association between depression with multiple measurements over time and colorectal cancer-specific mortality. In addition, further study should explore the interactions among depression, antidepressant use and colorectal cancer treatment on colorectal cancer-specific mortality.

**Table 3. Hazard ratios (HRs) and 95% confidence interval (CIs) for colorectal cancer outcome in relation to depression at WHI baseline stratified by tumor stage\*.**

| Variable | n | All-cause mortality | | | Colorectal cancer mortality | | |
|---|---|---|---|---|---|---|---|
| | | Deaths | HR | 95%CI | Deaths | HR | 95%CI |
| Tumor stage\*\* | | | | | | | |
| *Localized* | | | | | | | |
| Depression | | | | | | | |
| No | 855 | 306 | 1 | | 63 | 1 | |
| Yes | 156 | 55 | 1.00 | 0.75–1.35 | 10 | 0.82 | 0.41–1.64 |
| Depressive symptoms | | | | | | | |
| No | 901 | 320 | 1 | | 65 | 1 | |
| Yes | 110 | 41 | 1.05 | 0.75–1.48 | 8 | 0.87 | 0.41–1.87 |
| Antidepressant use | | | | | | | |
| No | 945 | 339 | 1 | | 71 | | |
| Yes | 66 | 22 | 0.93 | 0.60–1.45 | 2 | 0.47 | 0.11–1.98 |
| *Regional or distant* | | | | | | | |
| Depression | | | | | | | |
| No | 1,082 | 641 | 1 | | 431 | 1 | |
| Yes | 182 | 115 | 1.08 | 0.87–1.32 | 67 | 0.89 | 0.68–1.16 |
| Depressive symptoms | | | | | | | |
| No | 1,131 | 673 | 1 | | 448 | 1 | |
| Yes | 133 | 83 | 1.04 | 0.82–1.32 | 50 | 0.93 | 0.69–1.26 |
| Antidepressant use | | | | | | | |
| No | 1,199 | 717 | 1 | | 476 | | |
| Yes | 65 | 39 | 1.04 | 0.74–1.46 | 22 | 0.79 | 0.50–1.23 |

\* Adjusted for age at baseline, race/ethnicity, body mass index (BMI), smoking, physical activity, total energy intake, dietary fiber, percent calories from fat, comorbidity at baseline, family history of colorectal cancer mortality, receipt of colonoscopy, NSAIDs use, history of postmenopausal hormone use, and treatment arm in each CT.
\*\* The stage in some cases was unknown or in situ.

## Acknowledgments

Short list of WHI investigators.

Program Office: (National Heart, Lung, and Blood Institute, Bethesda, Maryland) Jacques Rossouw, Shari Ludlam, Joan McGowan, Leslie Ford, and Nancy Geller.

Clinical Coordinating Center: Clinical Coordinating Center: (Fred Hutchinson Cancer Research Center, Seattle, WA) Garnet Anderson, Ross Prentice, Andrea LaCroix, and Charles Kooperberg.

Investigators and Academic Centers: (Brigham and Women's Hospital, Harvard Medical School, Boston, MA) JoAnn E. Manson; (MedStar Health Research Institute/Howard University, Washington, DC) Barbara V. Howard; (Stanford Prevention Research Center, Stanford, CA) Marcia L. Stefanick; (The Ohio State University, Columbus, OH) Rebecca Jackson; (University of Arizona, Tucson/Phoenix, AZ) Cynthia A. Thomson; (University at Buffalo, Buffalo, NY) Jean Wactawski-Wende; (University of Florida, Gainesville/Jacksonville, FL) Marian Limacher; (University of Iowa, Iowa City/Davenport, IA) Jennifer Robinson; (University of Pittsburgh, Pittsburgh, PA) Lewis Kuller; (Wake Forest University School of Medicine, Winston-Salem, NC) Sally Shumaker; (University of Nevada, Reno, NV) Robert Brunner; (University of Minnesota, Minneapolis, MN) Karen L. Margolis.

Women's Health Initiative Memory Study: (Wake Forest University School of Medicine, Winston-Salem, NC) Mark Espeland.

## Author Contributions

**Conceptualization:** Xiaoyun Liang, Michael Hendryx, Juhua Luo.

**Data curation:** Xiaoyun Liang, Michael Hendryx, Juhua Luo.

**Formal analysis:** Xiaoyun Liang, Michael Hendryx, Lihong Qi, Dorothy Lane, Juhua Luo.

**Funding acquisition:** Juhua Luo.

**Writing – original draft:** Xiaoyun Liang.

**Writing – review & editing:** Xiaoyun Liang, Michael Hendryx, Lihong Qi, Dorothy Lane, Juhua Luo.

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
