## [Decision Letter · Decision Letter 0]

24 Sep 2020

PONE-D-20-10183

Effect of depression before colorectal cancer diagnosis on mortality among postmenopausal women

PLOS ONE

Dear Dr. Liang,

Thank you for submitting your manuscript to PLOS ONE. After careful consideration, we feel that it has merit but does not fully meet PLOS ONE’s publication criteria as it currently stands. Therefore, we invite you to submit a revised version of the manuscript that addresses the points raised during the review process.

The manuscript has been evaluated by two reviewers, and their comments are available below. You will see the reviewers have commented on the potential interest of your work. However, they have also raised a number of concerns that should be addressed before the manuscript can be accepted.

Please note that as per our publication criteria, PLOS ONE requires that all experiments, statistics and other analyses are performed to a high technical standard, described in sufficient detail and adhere to appropriate reporting guidelines and community standards. Conclusions must be presented in an appropriate fashion and be supported by the data (Please see http://journals.plos.org/plosone/s/criteria-for-publication). 

The reviewers raised concerns about the methods and presentation of the results. Please further clarify the timing of the baseline depression screening. Please also note the below requests to further describe the study sample. While the study sample arose from the previously published Women’s Health Initiative cohort, additional information describing the WHI cohort and the study sample included in this analysis should be added to the manuscript. Please also provide a rationale for considering race as a potential confounder, and clarify the operationalization of race as “white, non-Hispanic” or “other.”

We look forward to receiving your revised manuscript.

Kind regards,

Danielle Poole

Staff Editor

PLOS ONE

Journal Requirements:

2. Please note that PLOS does not allow reference to data not shown. Thus, before we proceed, we kindly ask you to provide the relevant data within the manuscript, the Supporting Information files, or in a public repository. If the data are not a core part of the research study being presented, please remove any references to these data.

Reviewers' comments:

Reviewer's Responses to Questions

**Comments to the Author**

1. Is the manuscript technically sound, and do the data support the conclusions?

Reviewer #1: Partly

Reviewer #2: Partly

2. Has the statistical analysis been performed appropriately and rigorously? 

Reviewer #1: I Don't Know

Reviewer #2: Yes

3. Have the authors made all data underlying the findings in their manuscript fully available?

Reviewer #1: Yes

Reviewer #2: Yes

4. Is the manuscript presented in an intelligible fashion and written in standard English?

Reviewer #1: Yes

Reviewer #2: Yes

5. Review Comments to the Author

Reviewer #1: Thank you for asking me to review this paper entitled: “Effect of depression before colorectal cancer diagnosis on mortality among postmenopausal women” by Liang et al. Using data from the Women’s Health Initiative (WHI) randomized trials and observational study, the authors evaluated the associations between depression prior to colorectal cancer (CRC) and CRC-specific and all-cause mortality.

My main concern with this study is that depression was ascertained only at a single point in time yet depression status changes over time. It would be different if the point in time was meaningful for clinical decision-making (e.g. depression at the time of CRC diagnosis), but in this study the authors are seeking to understand the potential role of pre-diagnosis depression on survival from CRC. Additional concerns are: 1) a lack of information about the elapsed time from ascertainment of depression to the start of follow-up, 2) a lack of information about the source population from which the cohort of women with colorectal cancer arose, and 3) the Discussion section is repetitive.

Title:

• The use of “effect” in the title implies the findings in this manuscript can be used to make a causal statement. “Association” would be more appropriate for this observational study.

Abstract:

• In the methods of the abstract, the authors state that depression was ascertained at “baseline”, however, they do not make clear that “baseline” refers to the definition of baseline in the WHI studies, rather than the start of follow-up in the present study.

• The tense in the first sentence of the conclusion should be past.

Introduction

• Lines 72-75: References are needed for each of the specific statements in this sentence, i.e., the specific studies that showed depressive symptoms were associated with an increased risk of CRC and the specific studies that showed that SSRIs were associated with a decreased risk of CRC.

• Lines 78-80: The direction of the association observed by Schofield et al. is not described.

• Lines 89-98: Other than biologic mechanisms, by what means may depression influence CRC survival? More background information about how socioeconomics, health care access, treatment adherence and comorbidities may lead to worse survival in CRC patients would be helpful.

• Lines 91-92: I suggest replacing “organism” with “a person”, if that is accurate.

Methods:

• Lines 106-111: More information is needed about the source population for this study, i.e., the WHI randomized trials and observational study. A reference to a prior publication is okay, but more information is needed in the manuscript itself. What were the inclusion and exclusion criteria in the WHI studies? How was postmenopausal defined?

• Line 108-111: What was the rationale for excluding women with CRC identified via death certificate only? Presumably one reason was that the date of diagnosis was unknown, but this is not stated. How many women were excluded for this reason?

• Line 112-114: What was the rationale for excluding women with a prior cancer diagnosis?

• Line 117-119: The statement about generating an indicator variable for persons with a missing family history needs to be clarified. Do the authors mean they included a missing category for the family history variable? If yes, all categories for this variable should be shown in Table 1 or the number with missing values described in a footnote. What was the rationale for creating a missing category (if that was done) for the family history covariate but not for the other covariates?

• Lines 122-127: More information about the ascertainment of the exposure is needed:

o Over what calendar years was the exposure ascertained in the WHI studies?

o Was the questionnaire mailed or in-person?

o When the authors refer to “baseline”, it’s not clear that they are referring to baseline of the WHI studies (as opposed to the start of follow-up in the present study).

• Line 146: How were CRC diagnoses ascertained? What data source was used?

• Line 148-150: The use of the term “censored observations” is not clear. It gives the impression that the record itself was censored rather than the follow-up time being censored.

• Line 152: The lowest age category was all women <70 years? That seems very broad. A description of the age of the source population (see above) may help the reader to better understand this grouping. The lower age limit should be specified.

• Line 153: Regarding the race variable:

o It seems unusual to capitalize “white”.

o Why were all races other than whites grouped together? Can they be shown separately in Table 1 even if they are grouped in the analyses?

o Can the authors provide a rationale for why they considered race a potential confounder in their analyses?

• Lines 151-164: What data sources were used for ascertainment of the covariates? What was the timing for ascertainment of the covariates? Also, some variables need to be defined more specifically (e.g. colonoscopy (ever/never?), physical activity and diet - over what time frame of the woman’s life, etc.)?

• Lines 158-161: What comorbidities were included in the comorbidity index? Here again, a reference to another paper is helpful, but some basic information is needed in the present manuscript.

• Lines 165-167: Duration of antidepressant use was a variable that was included in the models as an adjustment factor (e.g. lines 185-186). How was it possible to adjust for this variable if all unexposed women had the same value (i.e., 0)?

• Lines 167-169: This statement could be more clear. Does “not enrolled” refer to the observational study?

Statistical analysis

• Lines 177-181: A stratified Cox model was used to estimate the HR of all-cause mortality. Should a stratified model also have been used to estimate the HR for CRC specific mortality? Would this increase the comparability of the HRs estimated for each outcome (even though a stratified model may not have been necessary for CRC-specific mortality)?

• Line 179: Why did the authors adjust for age at CRC diagnosis instead of age at baseline in the WHI trial? Age at CRC diagnosis could be in a causal pathway if depression prior to diagnosis predisposes to CRC development at an older or younger age.

• Line 180: Tumor stage is also potentially in the causal pathway between depression prior to diagnosis and mortality if depression prior to diagnosis predisposes to diagnosis with low or high stage CRC.

• Line 185-186: See comment above about adjusting for duration of antidepressant use. How was this possible?

• Line 191-195: Please provide a brief rationale for about each of the sensitivity analyses.

o Please show the distribution of elapsed time from depression ascertainment to CRC diagnosis. This time is downstream of the depression itself, is there a concern that it could be in a causal pathway?

o How did the prevalence of depression change in the cohort when the cutpoint was changed (what was the number of women meeting this stricter definition)?

o It’s not totally clear what “Year 1” is referring to in the last sentence of this paragraph.

• Did this study need to be approved by the institution where the analyses were conducted? I understand that the WHI received IRB approval, but was approval also needed from the institution where these data were analyzed?

Results:

• Line 201-204: This sentence could be more clear. Some words seem to be missing (e.g. “more likely to be”…). Can the authors describe the races individually rather than group all races other than whites together?

• Line 214-216: Please describe the distribution of time from ascertainment of depression to CRC diagnosis. It would help to see this in a table. It would also be helpful to see the distribution of age at depression ascertainment.

• Please show the distribution of year of CRC diagnosis in Table 1.

Discussion:

• Line 224-225: This sentence is not worded clearly.

• Line 228-229: Please check reference #11, it appears to not be correct as it is a commentary that summarizes reference #10.

• Line 232-234: The wording in this sentence is odd because the authors state that their hypothesis is that an association exists, but they tested the hypothesis and their data did not support it.

• As mentioned above “baseline” is used throughout the paper but the timing of the baseline assessment relative to CRC diagnosis is not clear. One could easily confuse “baseline” with the start of follow-up in the present study.

• Line 243-247. If the authors had a concern about residual confounding by age, why did they not try to adjust for age more robustly in a post-hoc sensitivity analysis, e.g., examining the association among younger women only (and adjusting for age within that grouping (e.g. as a continuous variable))?

• Line 230-247: This section, describing the study limitations, appears to be redundant to some extent with the next paragraph (lines 249-264). These two paragraphs could be combined, and the study strengths could be a separate paragraph.

• Women in the intervention arms of the estrogen-only and estrogen plus progestin trials were randomized to receive menopausal hormones. Does use of menopausal hormones influence depression? How might this be a limitation in this study? How does adjustment for treatment arm (“not enrolled, intervention group, or control group”) address this limitation?

Table 1.

• Age – it’s not clear that the columns are percentages.

• BMI – appears to be percentages rather than “mean”.

• It would be helpful to see the number of women in each category rather than only percentages.

• Can all race categories be shown?

• Please show the distribution of year of CRC diagnosis.

Table 2.

• Using “deaths” rather than “cases” as a column heading would be more clear.

• In the title it’s not clear that baseline refers to the WHI baseline (same comment for Table 3).

Figure 1

• First box: what was the earliest year of ascertainment of cases?

Reviewer #2: Reviewer's report:

This is a paper that might be of interest for those in the related field. I have some comments and suggestions which may improve the quality of the manuscript.

1. Abstract

Background:” please remove repeated sentences the aim is so similar to the previous sentence; you should mention why it is important to study post-menopausal women?

Methods: time interval for the assessment of mortality is not clear

Results: it is confusing needs to be revised and please provide some statistical outputs

2. Introduction

Why is this topic important in women? Only the availability of the data does not support generating a research question it should be defined scientifically why this research question is the matter among women, any supportive evidence that they are more likely to suffer depression and so on

3. Results

Authors refer the reader to the Tables; they need to provide some key results and measures in the text.

General comments

Please remove the post-menopausal from the topic since there is no information or results or supportive text about this

The introduction should be revised

Results section are not clear

There is no information about depression symptoms following the colorectal cancer diagnosis; it was important to consider this as a part of the analysis

6. PLOS authors have the option to publish the peer review history of their article (what does this mean?). If published, this will include your full peer review and any attached files.

Reviewer #1: No

Reviewer #2: No

---

## [Author Response · Author response to Decision Letter 0]

12 Nov 2020

The 1st reviewer

Thank you for asking me to review this paper entitled: “Effect of depression before colorectal cancer diagnosis on mortality among postmenopausal women” by Liang et al. Using data from the Women’s Health Initiative (WHI) randomized trials and observational study, the authors evaluated the associations between depression prior to colorectal cancer (CRC) and CRC-specific and all-cause mortality. 

My main concern with this study is that depression was ascertained only at a single point in time yet depression status changes over time. It would be different if the point in time was meaningful for clinical decision-making (e.g. depression at the time of CRC diagnosis), but in this study the authors are seeking to understand the potential role of pre-diagnosis depression on survival from CRC. Additional concerns are: 1) a lack of information about the elapsed time from ascertainment of depression to the start of follow-up, 2) a lack of information about the source population from which the cohort of women with colorectal cancer arose, and 3) the Discussion section is repetitive. 

It is a study limitation that depression was ascertained only at WHI baseline. We have acknowledged the limitation in the Discussion.

We added information about duration of depression at WHI baseline to colorectal cancer diagnosis. Please see No. 25, 29 and 41 comments.

We added information about the source population. Please see No. 8 comment.

We combined and re-organized the last two paragraphs in the Discussion. Please see No. 35 comment.

Title

1. The use of “effect” in the title implies the findings in this manuscript can be used to make a causal statement. “Association” would be more appropriate for this observational study. 

The manuscript title has been revised as suggested.

Abstract:

2. In the methods of the abstract, the authors state that depression was ascertained at “baseline”, however, they do not make clear that “baseline” refers to the definition of baseline in the WHI studies, rather than the start of follow-up in the present study. 

An explanation was added in the methods of the Abstract: Baseline means screening visit in the WHI study.

3. The tense in the first sentence of the conclusion should be past.

The tense in the first sentence of the conclusion has been revised as suggested.

Introduction

4. Lines 72-75: References are needed for each of the specific statements in this sentence, i.e., the specific studies that showed depressive symptoms were associated with an increased risk of CRC and the specific studies that showed that SSRIs were associated with a decreased risk of CRC. 

For each of the two statements, one reference was added.

5. Lines 78-80: The direction of the association observed by Schofield et al. is not described. 

Information about the direction of the association was added.

6. Lines 89-98: Other than biologic mechanisms, by what means may depression influence CRC survival? More background information about how socioeconomics, health care access, treatment adherence and comorbidities may lead to worse survival in CRC patients would be helpful. 

More background information was added: A second possible mechanism is that people with depression might change behaviors (such as engaging in less physical activity and more smoking), be less likely to engage in social activity (resulting in less social support and social capital), or have less adherence to medical service (such as preventive screening and medical treatment), which could lead to worse survival in cancer patients.

7. Lines 91-92: I suggest replacing “organism” with “a person”, if that is accurate. 

Replacement has been done as suggested.

Methods:

8. Lines 106-111: More information is needed about the source population for this study, i.e., the WHI randomized trials and observational study. A reference to a prior publication is okay, but more information is needed in the manuscript itself. What were the inclusion and exclusion criteria in the WHI studies? How was postmenopausal defined? 

Two references (The Women's Health Initiative Study Group. Design of the Women's Health Initiative clinical trial and observational study. Control Clin Trials. 1998;19:61-109) (Hays J, Hunt JR, Hubbell FA, Anderson GL, Limacher M, Allen C, et al. The Women's Health Initiative recruitment methods and results. Ann Epidemiol 2003;13:S18-S77) have been added.

Inclusion and exclusion criteria were added.

The definition of postmenopausal was added: Women aged 50–54 with no menstrual period for at least one year and women aged 55 or above with no menstrual period for at least half a year.

9. Line 108-111: What was the rationale for excluding women with CRC identified via death certificate only? Presumably one reason was that the date of diagnosis was unknown, but this is not stated. How many women were excluded for this reason?

Of 3217 women with colorectal cancer, 2994 centrally adjudicated were included in the current study, and 223 determined solely by the cause of death without central adjudication were excluded from the study because the date of cancer diagnosis was unknown, which was added in the Participants of Methods.

10. Line 112-114: What was the rationale for excluding women with a prior cancer diagnosis? 

A prior cancer diagnosis may likely result in postdiagnosis depression, and if it was not excluded, we could not judge whether the depression at WHI baseline is prediagnotic or postdiagnostic.

11. Line 117-119: The statement about generating an indicator variable for persons with a missing family history needs to be clarified. Do the authors mean they included a missing category for the family history variable? If yes, all categories for this variable should be shown in Table 1 or the number with missing values described in a footnote. What was the rationale for creating a missing category (if that was done) for the family history covariate but not for the other covariates?

After women with missing data for continuous covariates (physical activity, total energy intake, dietary fiber, and percent calories from fat) were excluded (n=134), we also excluded a few women with missing data for categorical covariates (smoking (n=30), body mass index (n=23), or receipt of colonoscopy (n=13)). However, many subjects were missing for family history (n=193), and we would lose more information if we dropped them, so it is suitable to create an indicator variable for this covariate. The number with missing values for CRC family history was described in the footnote of Table 1.

12. Lines 122-127: More information about the ascertainment of the exposure is needed:

o Over what calendar years was the exposure ascertained in the WHI studies?

o Was the questionnaire mailed or in-person?

o When the authors refer to “baseline”, it’s not clear that they are referring to baseline of the WHI studies (as opposed to the start of follow-up in the present study). 

The exposure was ascertained during 1993-1998, which was added in the Exposures of Methods.

The questionnaire was administered in the face-to-face interview, which was added in the Exposures of Methods.

An explanation was added in the Participants of Methods: “Baseline” in the current study refers to the screening visit in the WHI study.

13. Line 146: How were CRC diagnoses ascertained? What data source was used?

A sentence was added in the Outcomes of Methods: The diagnosis of colorectal cancer was identified via self-reported questionnaires, verified by physicians with medical records and pathology reports, and locally adjudicated by a Clinical Center and then centrally by the WHI Clinical Coordinating Center.

14. Line 148-150: The use of the term “censored observations” is not clear. It gives the impression that the record itself was censored rather than the follow-up time being censored.

“Censored observations” was rephrased as “administratively censored”.

15. Line 152: The lowest age category was all women <70 years? That seems very broad. A description of the age of the source population (see above) may help the reader to better understand this grouping. The lower age limit should be specified.

In the current study women at the CRC diagnosis aged <70, 70-80 and ≥80 accounted for 32%, 45% and 23%, respectively. Women aged 50 to 79 years old were enrolled in the WHI study, which was added in the Participants of Methods, and were diagnosed with colorectal cancer after a mean of 8.8 years, so they would be 8.8 years older at average at the diagnosis of colorectal cancer than at the baseline. We have replaced age at diagnosis by age at baseline in the models.

16. Line 153: Regarding the race variable:

o It seems unusual to capitalize “white”.

o Why were all races other than whites grouped together? Can they be shown separately in Table 1 even if they are grouped in the analyses?

o Can the authors provide a rationale for why they considered race a potential confounder in their analyses?

“White” was changed as lowercase.

White women accounted for 84.6% of all participants, so we grouped other races together. Proportions of American Indian or Alaskan Native, Asian or Pacific Islander, Black or African-American, Hispanic or Latino, non-Hispanic-white were shown separately in Table 1.

Some literature showed that persons with different races had various level of mortality, and our data supported that race was associated with depression so we considered race as a potential confounder in the analysis. We have provided the rationale and related literatures in the Covariates of Methods. 

17. Lines 151-164: What data sources were used for ascertainment of the covariates? What was the timing for ascertainment of the covariates? Also, some variables need to be defined more specifically (e.g. colonoscopy (ever/never?), physical activity and diet - over what time frame of the woman’s life, etc.)? 

Data sources and timing about covariates ascertainment were added: Baseline covariates in WHI study, including demographic and lifestyle data as well as data about family and medical history, were acquired by self-reported questionnaires during 1993-1998.

Classification of colonoscopy was changed into ever and never.

Time frames of physical activity and diet measurement were added.

18. Lines 158-161: What comorbidities were included in the comorbidity index? Here again, a reference to another paper is helpful, but some basic information is needed in the present manuscript.

Information about comorbidities was added: non-cancer chronic diseases such as glaucoma, asthma, cardiovascular disease and hypertension.

19. Lines 165-167: Duration of antidepressant use was a variable that was included in the models as an adjustment factor (e.g. lines 185-186). How was it possible to adjust for this variable if all unexposed women had the same value (i.e., 0)? 

Duration of antidepressant use was removed from the models.

20. Lines 167-169: This statement could be more clear. Does “not enrolled” refer to the observational study? 

”Not enrolled” does not exactly refer to the observational study. WHI has multiple CTs, not all women who were not enrolled into a specific trial are OS.

Statistical analysis

21. Lines 177-181: A stratified Cox model was used to estimate the HR of all-cause mortality. Should a stratified model also have been used to estimate the HR for CRC specific mortality? Would this increase the comparability of the HRs estimated for each outcome (even though a stratified model may not have been necessary for CRC-specific mortality)? 

A stratified model was carried out to estimate the HR for CRC specific mortality, and the results were almost the same as that estimated in a Cox model.

22. Line 179: Why did the authors adjust for age at CRC diagnosis instead of age at baseline in the WHI trial? Age at CRC diagnosis could be in a causal pathway if depression prior to diagnosis predisposes to CRC development at an older or younger age.

Age at baseline was included in all the models instead of age at diagnosis.

23. Line 180: Tumor stage is also potentially in the causal pathway between depression prior to diagnosis and mortality if depression prior to diagnosis predisposes to diagnosis with low or high stage CRC. 

We found that results did not change with and without adjusting for tumor stage in the models.

24. Line 185-186: See comment above about adjusting for duration of antidepressant use. How was this possible?

Duration of antidepressant use was removed from the models.

25. Line 191-195: Please provide a brief rationale for about each of the sensitivity analyses. 

o Please show the distribution of elapsed time from depression ascertainment to CRC diagnosis. This time is downstream of the depression itself, is there a concern that it could be in a causal pathway?

o How did the prevalence of depression change in the cohort when the cutpoint was changed (what was the number of women meeting this stricter definition)? 

o It’s not totally clear what “Year 1” is referring to in the last sentence of this paragraph. 

Duration from WHI baseline to colorectal cancer diagnosis was added in the Results and Table 1. We found that results did not change with and without adjusting for duration from baseline to colorectal cancer diagnosis in the models.

When depressive symptoms was categorized as no/yes based on a cutpoint of 0.06 and 0.009, its prevalence was 10.8% (n=259) and 23.6% (n=565), respectively.

Year 1 was replaced as one-year follow up.

26. Did this study need to be approved by the institution where the analyses were conducted? I understand that the WHI received IRB approval, but was approval also needed from the institution where these data were analyzed?

The approval was not needed from the institution where de-identified data were analyzed.

Results:

27. Line 201-204: This sentence could be more clear. Some words seem to be missing (e.g. “more likely to be”…). Can the authors describe the races individually rather than group all races other than whites together? 

“More likely to be ” was added. Proportions of American Indian or Alaskan Native, Asian or Pacific Islander, Black or African-American, Hispanic or Latino, non-Hispanic-white were shown separately In Table 1, and we could see that women with depression at baseline were more likely to be black compared to those without depression. Description about duration from baseline to colorectal cancer diagnosis was added in the first paragraph of Results.

28. Line 214-216: Please describe the distribution of time from ascertainment of depression to CRC diagnosis. It would help to see this in a table. It would also be helpful to see the distribution of age at depression ascertainment.

Duration from WHI baseline to colorectal cancer diagnosis was added in the Results and Table 1. Age at baseline was also added in Table 1.

29. Please show the distribution of year of CRC diagnosis in Table 1.

Duration from WHI baseline to colorectal cancer diagnosis was added in Table 1.

Discussion

30. Line 224-225: This sentence is not worded clearly. 

We removed this sentence, because in Satin et al’s study, depression was measured after colorectal cancer diagnosis, but in the current study we measured depression before colorectal cancer diagnosis.

31. Line 228-229: Please check reference #11, it appears to not be correct as it is a commentary that summarizes reference #10.

We have removed the incorrect reference.

32. Line 232-234: The wording in this sentence is odd because the authors state that their hypothesis is that an association exists, but they tested the hypothesis and their data did not support it. 

The statement “assuming our hypothesis is correct that there is an association between depression and colorectal cancer prognosis” was removed.

33. As mentioned above “baseline” is used throughout the paper but the timing of the baseline assessment relative to CRC diagnosis is not clear. One could easily confuse “baseline” with the start of follow-up in the present study. 

An explanation was added in the Methods: “Baseline” in the current study refers to the screening visit in the WHI study.

34. Line 243-247. If the authors had a concern about residual confounding by age, why did they not try to adjust for age more robustly in a post-hoc sensitivity analysis, e.g., examining the association among younger women only (and adjusting for age within that grouping (e.g. as a continuous variable))? 

We carried out an analysis among women under 60 at baseline, and similar results were found as that in the main analysis, which was added in the manuscript where we discussed residual confounding by age. 

35. Line 230-247: This section, describing the study limitations, appears to be redundant to some extent with the next paragraph (lines 249-264). These two paragraphs could be combined, and the study strengths could be a separate paragraph. 

These two paragraphs were combined and re-organized. The first paragraph describes the limits of the depression measure, and the second paragraph describes the study strengths and another limitation.

36. Women in the intervention arms of the estrogen-only and estrogen plus progestin trials were randomized to receive menopausal hormones. Does use of menopausal hormones influence depression? How might this be a limitation in this study? How does adjustment for treatment arm (“not enrolled, intervention group, or control group”) address this limitation? 

Yes, menopausal hormone use may influence depression. However, in the Cox models, we adjusted for treatment arm in each clinical trial. For Hormone Replacement Therapy trial, we included five arms: estrogen-alone intervention, estrogen-alone control, estrogen plus progestin intervention, estrogen plus progestin control, and not enrolled. In addition, history of postmenopausal hormones use was also added in the models as a covariate, and its distribution was shown in Table 1.

Table 1.

37. Age – it’s not clear that the columns are percentages.

Percentages were added as suggested.

38. BMI – appears to be percentages rather than “mean”.

Replacement was made as suggested.

39. It would be helpful to see the number of women in each category rather than only percentages.

Number of women in each category was added.

40. Can all race categories be shown?

Proportions of American Indian or Alaskan Native, Asian or Pacific Islander, Black or African-American, Hispanic or Latino, non-Hispanic-white were shown separately.

41. Please show the distribution of year of CRC diagnosis.

Distribution of duration from WHI baseline to CRC diganosis was added.

Two categories of variable were grouped in Table 1 including WHI baseline characteristics and characteristics at CRC diagnosis.

Table 2.

42. Using “deaths” rather than “cases” as a column heading would be more clear.

“Cases” has been replaced by “deaths” as suggested.

43. In the title it’s not clear that baseline refers to the WHI baseline (same comment for Table 3).

In the titles of Table 2 and Table 3, “WHI” was added before “baseline”.

Figure 1

44. First box: what was the earliest year of ascertainment of cases?

The earliest time of CRC diagnosis was 4 days since enrollment. 

The 2nd reviewer

Title: Effect of depression before colorectal cancer diagnosis on mortality among postmenopausal women

Date: 15/09/2020

Reviewer's report:

This is a paper that might be of interest for those in the related field. I have some comments and suggestions which may improve the quality of the manuscript. 

Abstract

1. Background:” please remove repeated sentences the aim is so similar to the previous sentence; you should mention why it is important to study post-menopausal women?

Repeated sentences were revised as suggested. WHI is a study among postmenopausal women, and in the manuscript we added the definition of postmenopausal in the Participants of Methods and women’s more suffering from depression in the last paragraph of Introduction.

2. Methods: time interval for the assessment of mortality is not clear

Time interval was added as suggested.

3. Results: it is confusing needs to be revised and please provide some statistical outputs 

HRs and 95% confidence intervals were added for associations between baseline depression and all-cause mortality and colorectal cancer-specific mortality.

Introduction

4. Why is this topic important in women? Only the availability of the data does not support generating a research question it should be defined scientifically why this research question is the matter among women, any supportive evidence that they are more likely to suffer depression and so on

The statement “The prevalence of depression among women is approximately twice that of men” and cited literature was added in the last paragraph of Introduction.

Results

5. Authors refer the reader to the Tables; they need to provide some key results and measures in the text. 

HRs and 95% confidence intervals were added for associations between baseline depression and all-cause mortality and colorectal cancer-specific mortality.

General comments

6. Please remove the post-menopausal from the topic since there is no information or results or supportive text about this 

WHI is a study among postmenopausal women, and in the manuscript we added the definition of postmenopausal in the Participants of Methods.

7. The introduction should be revised 

We have made a number of revisions to the Introduction to describe the background and study purpose more clearly.

8. Results section are not clear 

We have modified the Results section in response to multiple comment from reviewers.

9. There is no information about depression symptoms following the colorectal cancer diagnosis; it was important to consider this as a part of the analysis 

In the current study, depressive symptoms and antidepressant use only at baseline were considered, and we could not get the information about depression symptoms following colorectal cancer diagnosis, which is a study limitation mentioned in the Discussion.

Level of interest: this might be of interest for researchers in this field

Statistical review: I have checked and made some suggestions

---

## [Editor Report · Decision Letter 1]

16 Dec 2020

Association between prediagnosis depression and mortality among postmenopausal women with colorectal cancer

PONE-D-20-10183R1

Dear Dr. Liang,

We’re pleased to inform you that your manuscript has been judged scientifically suitable for publication and will be formally accepted for publication once it meets all outstanding technical requirements.

Kind regards,

Yiqiang Zhan

Academic Editor

PLOS ONE
---

## [Editor Report · Acceptance letter]

18 Dec 2020

PONE-D-20-10183R1 

Association between prediagnosis depression and mortality among postmenopausal women with colorectal cancer 

Dear Dr. Liang:

I'm pleased to inform you that your manuscript has been deemed suitable for publication in PLOS ONE. Congratulations! Your manuscript is now with our production department. 

Kind regards, 

on behalf of

Dr. Yiqiang Zhan 

Academic Editor

PLOS ONE